# BCNet: A Deep Learning Computer-Aided Diagnosis Framework for Human Peripheral Blood Cell Identification

**DOI:** 10.3390/diagnostics12112815

**Published:** 2022-11-16

**Authors:** Channabasava Chola, Abdullah Y. Muaad, Md Belal Bin Heyat, J. V. Bibal Benifa, Wadeea R. Naji, K. Hemachandran, Noha F. Mahmoud, Nagwan Abdel Samee, Mugahed A. Al-Antari, Yasser M. Kadah, Tae-Seong Kim

**Affiliations:** 1Department of Electronics and Information Convergence Engineering, College of Electronics and Information, Kyung Hee University, Suwon-si 17104, Republic of Korea; 2Department of Studies in Computer Science, University of Mysore, Manasagangothri, Mysore 570006, India; 3IoT Research Center, College of Computer Science and Software Engineering, Shenzhen University, Shenzhen 518060, China; 4Centre for VLSI and Embedded System Technologies, International Institute of Information Technology, Hyderabad 500032, India; 5Department of Science and Engineering, Novel Global Community Educational Foundation, Hebersham, NSW 2770, Australia; 6Department of Computer Science and Engineering, Indian Institute of Information Technology Kottayam, Kerala 686635, India; 7Department of Artificial Intelligence, Woxsen University, Hyderabad 502345, India; 8Rehabilitation Sciences Department, Health and Rehabilitation Sciences College, Princess Nourah bint Abdulrahman University, Riyadh 11671, Saudi Arabia; 9Department of Information Technology, College of Computer and Information Sciences, Princess Nourah bint Abdulrahman University, Riyadh 11671, Saudi Arabia; 10Department of Artificial Intelligence, College of Software and Convergence Technology, Daeyang AI Center, Sejong University, Seoul 05006, Republic of Korea; 11Electrical and Computer Engineering Department, King Abdulaziz University, Jeddah 22254, Saudi Arabia; 12Biomedical Engineering Department, Cairo University, Giza 12613, Egypt

**Keywords:** blood cell, new BCNet framework, deep transfer learning, multi-class identification, verification and validation

## Abstract

Blood cells carry important information that can be used to represent a person’s current state of health. The identification of different types of blood cells in a timely and precise manner is essential to cutting the infection risks that people face on a daily basis. The BCNet is an artificial intelligence (AI)-based deep learning (DL) framework that was proposed based on the capability of transfer learning with a convolutional neural network to rapidly and automatically identify the blood cells in an eight-class identification scenario: Basophil, Eosinophil, Erythroblast, Immature Granulocytes, Lymphocyte, Monocyte, Neutrophil, and Platelet. For the purpose of establishing the dependability and viability of BCNet, exhaustive experiments consisting of five-fold cross-validation tests are carried out. Using the transfer learning strategy, we conducted in-depth comprehensive experiments on the proposed BCNet’s architecture and test it with three optimizers of ADAM, RMSprop (RMSP), and stochastic gradient descent (SGD). Meanwhile, the performance of the proposed BCNet is directly compared using the same dataset with the state-of-the-art deep learning models of DensNet, ResNet, Inception, and MobileNet. When employing the different optimizers, the BCNet framework demonstrated better classification performance with ADAM and RMSP optimizers. The best evaluation performance was achieved using the RMSP optimizer in terms of 98.51% accuracy and 96.24% F1-score. Compared with the baseline model, the BCNet clearly improved the prediction accuracy performance 1.94%, 3.33%, and 1.65% using the optimizers of ADAM, RMSP, and SGD, respectively. The proposed BCNet model outperformed the AI models of DenseNet, ResNet, Inception, and MobileNet in terms of the testing time of a single blood cell image by 10.98, 4.26, 2.03, and 0.21 msec. In comparison to the most recent deep learning models, the BCNet model could be able to generate encouraging outcomes. It is essential for the advancement of healthcare facilities to have such a recognition rate improving the detection performance of the blood cells.

## 1. Introduction

Blood cancer, lung cancer, breast cancer, and colon cancer are just a few of the many disorders for which microscopic image analysis plays a crucial role in diagnosis and early detection [1,2]. Leukemia is another term for cancer of the blood [3]. This malignancy begins in the bone marrow and manifests as an abnormal number or shape of white blood cells (leucocytes) in the blood [4,5,6]. In 2020, it was estimated that 19.3 million people have been diagnosed with cancer (18.1 million without nonmelanoma skin cancer) and that nearly 10 million people would lose their lives to the disease [7]. Indeed, cancer is responsible for one out of every six deaths around the world, especially in developing nations, where cancer accounts for a staggering 70% of all fatalities [8]. It is anticipated that there will be 1,918,030 new cancer cases and 609,360 cancer-related fatalities in the United States in the year 2022. Out of these deaths, about 350 each day will be attributable to lung cancer, which is the leading cause of cancer-related mortality. The Cancer Statistics Center predicts that there will be 60,650 new cases of acute leukemia and 24,000 deaths from the disease in 2022, with an incidence rate of 14.2 per 100,000 people and a mortality rate of 6.1 per 100,000 people over that time period [9]. Leukemia is a potentially fatal disease that can be slowed or stopped with early treatment. As a result, the prompt diagnosis and treatment of Leukemia is urgently required [10,11]. Hematological diagnosis relies heavily on microscopic examination and the cell classification of blood [12,13]. Diagnosis of hematological malignancies such as Acute Myeloid Leukemia (AML) begins with a morphological analysis of leukocytes either from peripheral blood or the bone marrow [14] Cytomorphology is especially important in the standard French-American-British (FAB) categorization of acute myeloid leukemias [15]. The cytomorphological analysis of leukocytes is a standard aspect of hematological diagnostic workup that has so far resisted automation and is typically carried out by educated human professionals. Because of the high degree of intra/inter-observer variation that is difficult to account for and the scarcity of appropriately qualified experts, cytomorphological classification is a laborious and time-consuming process [8].

Medical experts rely on the medical imaging modalities such as computed tomography (CT), microscopic blood smear images, Magnetic Resonance Imaging (MRI), X-ray, and ultrasound (US) to diagnose health challenges and assign treatment prescriptions [16,17]. Researchers and developers are able to deliver smart solutions for medical imaging diagnoses thanks to the AI-based potential functionalities of machine learning and deep learning technologies [18,19,20,21,22,23]. Recently, several smart Computer-Aided Diagnosis (CAD) models have been created to handle automatic medical diagnosis purposes. This is to support the medical staff in performing a fast and accurate examination and diagnosis results, especially during the epidemic or pandemic health situation. The haematological examinations are a daily routine in any hospital or medical centre. The role of deep learning CAD system is important in terms of automatically identifying the variety of the patient’s haematological conditions with blood cancer or leukaemia. The microscopic blood smear imagining technique is used for diagnosing leukemia by analyzing and identifying the different blood cells [24]. In the pathology lab assessments, white blood cells’ (WBC) classification and identification subclasses is attained. In addition, the lab examination of leukocytes includes monocytes, lymphocytes, neutrophils, basophils, and eosinophils. To identify the different types of blood cells in a rapid way and fulfil the physician’s requirements is a big challenge. Thus, AI technologies should interact and contribute by providing smart examination solutions without user intervention. The need of AI smart solutions motivates us to develop a novel schema of the CAD system based on the new BCNet deep learning model.

Recently, various investigations on blood cell classification using ML and Deep Learning have been carried out [3,4,5,6,25,26,27,28].There are a number of different optimization algorithms that may be used for training deep neural networks, and the one that is chosen can have an effect on how well the DL model functions. Many hyper-parameters can be adjusted to improve the neural network’s performance. Every one of these factors affects network speed in some way, but not all do so to the same extent. How well the algorithm converges to a solution, or whether it explodes, could depend on the optimizer that has been selected. Among the available optimizers are ADAM, RMSP, and SGDM. However, the great majority of recent image classification efforts using the DL approach have relied on the ADAM optimizer. To the best of our knowledge, no prior research has investigated the impact of competing optimizers on DL’s efficacy. The main contributions of this study are briefly summarized here:A deep learning schema of the CAD system based on the newly deep learning BCNet is proposed in order to identify multiclass blood cells rapidly and automatically.The multiple class identification task is conducted in terms of improving the overall classification performance.A comprehensive evaluation experiment is conducted to investigate the reliability and feasibility of the proposed BCNet using multiple optimizers and different state-of-the-art deep learning models such as DensNet, ResNet, Inception, and MobileNet.

The rest of the paper is organized in the following manner. Section 2 introduces the survey of the latest related works. Section 3 explains in detail the technical schema of the proposed BCNet. The assessment performance results are explained and discussed in Section 4 and Section 5, respectively. The findings of this study are discussed in the conclusion section.

## 2. Related Works

Several artificial intelligence models have been employed for over two decades to recognize the different types of blood cells. Red blood cells (RBCs), platelets (PRBCs), and white blood cells (WBCs) are the three basic types of blood cells; all three play important roles in the human immune response [29,30,31].

Medical professionals can benefit from the use of ML and DL methods for WBC categorization because it requires less work on their part and yields faster, more accurate results. Variations in the ratio of neutrophils, eosinophils, basophils, monocytes, and lymphocytes between healthy and diseased patients are readily apparent and play a significant role in diagnosis [32]. The combinatory approach of machine learning and the deep learning-based approach for the classification of WBC images were able to achieve 97.57% accuracy [33]. Changhun et al. proposed a W-Net model in a combination of CNN with RNN with DCGANs for image synthesizing later used for WBC classification, and attained an accuracy of 97% for a 5 class dataset [34]. César Cheuque et al. proposed the MLCNN detection of white blood cell Faster RCNN used to extract Region of interest later with Mobilenet based model is used to train the classification framework gained performance accuracy of 98.4% [35]. In continuation Next BCNet [36] to address the blood cell classification for three classes via transfer learning approach with ResNet18 as backbone model for learning and noted 96.78% accuracy. A deep learning based AI framework artificial intelligence-based microscopy image classifier for blood cell classification is proposed with transfer learning methods and realized an accuracy of 98.6% [37]. Furthermore, Acevedo et al. in [32] used the HPBC dataset to build their deep learning framework for classification of blood cells. Classifying different types of blood cells is an important activity that aids hematologists in making intelligent treatment decisions. Using the BCCD datasets, which are four-class datasets, Liang et al. presented a combination of RNN and CNN models to classify blood cells. These models included Inception-v3, ResNet-50, and Xception networks, and were then merged with an LSTM block to achieve the classification [38]. A transfer learning strategy was proposed by Acevedo et al. with the purpose of automating the classification of blood cells. They have utilized the VGG-16 and Inception-V3 as feature extractors in conjunction with the SVM and SoftMax as classifiers for the eight different types of balanced and imbalanced human peripheral blood cells (HPBC). They obtained the data using the CellaVision equipment, and they determined that the classification accuracy was 96.2% [32]. For the purpose of classifying WBC, Almezhghwi et al. have implemented several distinct kinds of Deep CNN models, including VGG, ResNet, and DenseNet, on the datasets provided by LSIC. Neutrophils, eosinophils, lymphocytes, monocytes, and basophils are the several types of white blood cells that make up the WBC. Data augmentation is accomplished by the utilization of transformation strategies, and in addition to that, they have implemented GANs for the purpose of producing datasets. They enhanced their accuracy to 98.8% and attained this result [39]. To categorize WBC [40], it is common practice to first segment the cells, extract defining properties, and then classify them. Such a method relies heavily on proper segmentation and has a poor degree of accuracy. Additionally, the precision of classification models is affected by insufficient datasets or unbalanced classes for related samples when using deep learning for medical diagnoses. To deal with such tasks, the authors in [40] proposed using a Deep Convolutional Generative Adversarial Network (DC-GAN) or a Residual neural network (ResNet). The experimental results show that the model has a high accuracy (91.7%) when it comes to classifying WBC images. Inadequate data samples and the uneven distribution of classes are two problems that this model addresses and corrects. When compared to other network architectures, the proposed technique provides the highest accuracy for classifying WBC images [40,41,42]. Wijesinghe et al. [43] have also presented a new method for the classification of WBC images. K-means cluster analysis using RGB color components and manual thresholding is used to determine the region of interest. The manually cropped WBC images then classified them using a VGG-16-based method. On custom data, they achieved a 97.4% detection rate and a 95.89% classification rate, respectively. Classification, localization, and detection of Leukocytes were proposed by Zhao et al. [44] Databases from Cellavision and Jiashan have been used for detection, while ALL-IDB has been used for classification. Pipelining begins with WBC identification using morphological methods, then moves on to color and granularity features for classification. An SVM Classifier has been used to classify the data as either basophil or eosinophil. On the other hand, a CNN/random forest hybrid model was used to classify the remaining classes, which included neutrophils, lymphocytes, and monocytes. In the end, 92.8% accuracy was achieved. Using DL models including ResNet, AlexNet, GoogLeNet, ZFNet, VGGNet, DenseNet, and Cell3Net, Qin et al. [29] suggested a classification technique for leukocytes with 40 classes. For all 40 courses, the achieved accuracy was 76.84 percent. In continuation, Zheng et al. [45] have proposed a self-supervised approach to blood cell segmentation. To extract the foreground from the background, they used an unsupervised technique based on K-means clustering. After that, they utilized a supervised method to extract features from RGB and HSV colors. Classification was then performed using the SVM on data from CellaVision and the Jiangxi Tecom Science Corporation of China. Their error rates were 3.18 and 0.69 for the CellaVision and Jiangxi Tecom Science Corporation and China datasets, respectively. Sajjad et al. have introduced a mobile-cloud-based model to handle segmentation followed by the classification of leukocytes into their five separate groups, as explained in [46]. Blood cells were distinguished using K-means and morphological procedures. In all, 1030 WBC blood smear samples were given by Hayatabad Medical Center (HMC). Segmented regions are analyzed for texture, geometry, and statistical data, then compared using different classifiers as EMC SVM. The segmentation strategy has a nucleus accuracy of 95.7%, a cytoplasm accuracy of 91.3%, and a classification accuracy of 98.6%.

## 3. Materials and Methods

In this section, we discuss the technical aspects of our work, including the dataset definition, the proposed BCNet architecture, and the components that make up the BCNet.

### 3.1. Dataset

A public dataset of human peripheral blood cells (also known as HPBC) is utilized in the process of constructing and assessing various models, including the deep learning BCNet. The blood cell images were created by Anna et al. using data that was acquired from the medical clinic in Barcelona between the years 2015 and 2019. [32]. The data repository consists a total of 17,092 images of blood cells of healthy individuals. The dataset is publicly available via https://data.mendeley.com/datasets/snkd93bnjr/1 (accessed on 21 June 2022).

The blood cell images are extracted from the whole microscopic scan slides involving eight subcategories, which are Basophil (BAS), Eosinophil (EOS), Erythroblast (ERY), Immature Granulocytes (IMM), Lymphocyte (LYM), Monocyte (MON), Neutrophil (NEU), and Platelet (PLT). All blood cell images are in the RGB color space in the ‘jpg’ format and with a size of 360 × 360 pixels. Each image was annotated by the pathologists of the hospital clinic of Barcelona. Captured images were obtained from individuals without infections hematologic or oncologic disease and during blood sample collection individuals have not undergone any pharmacologic treatments. Some examples of blood cell images per class are depicted in Figure 1.

Moreover, the data distribution of all images used in this study is summarized over all classes, as demonstrated in Figure 2.

### 3.2. Pre-Processing

The initial step in the image classification procedure, depicted in Figure 3, involves the pre-processing of blood cell images. It is used to improve the proposed BCNet’s overall performance by cleaning and preparing the images. Bi-cubic interpolation is used to scale all images to a final resolution of 224 × 224 pixels. Meanwhile, normalization of intensity is performed so that all pixel values fall inside the range [0, 255]. The augmentation strategy of the training set is performed to enlarge the dataset size and improve the overall performances, as performed in our previous works [47,48].

### 3.3. Data Preparation: Training, Validation, and Testing

To fine-tune and assess the proposed BCNet, the images per class are divided into three categories: the training set (70%), validation set (10%), and testing set (20%). Figure 2 depicts the number of blood cell images per category for each class. The trainable parameters such as bias and weights of the proposed CNN model are optimized via the training process by utilizing the training–validation sets. Subsequently, the overall performance of the proposed deep learning model is assessed by an evaluation set. Furthermore, the proposed BCNet model is evaluated through five-fold tests for training, validation, and evaluation sets. To deliver a robust and feasible CAD system for the detection and classification of HPBC data into eight classes, the k-fold cross-validation strategy plays an important role in the case of lower datasets per each class.

### 3.4. The Proposed Deep Learning Framework

The BCNet has been developed to tackle the blood cell classification problem and to improve the overall identification performance as shown in Figure 3. It can be extended and generalized to be used for different domains such as biomedical imaging with different modalities such as X-ray, MRI, CT, US, microscopic imaging, satellite imaging, etc. In the training process, the deep learning BCNet model has been adopted based on different key factors of the width and depth of the network that can be seen in the previous ones: ResNet [49] and WideResNet [50]. Such factors are helpful for training the models and attaining a higher identification performance. To build and develop the BCNet, we used the EfficientNet model as a base network, which is adopted based on the fundamental blocks of mobile inverted bottleneck convolution (MBConv) from MobileNet [51]. In [38], the EfficientNet model proposed the fundamental relationship, which explains the width and depth parameters’ efficacy for better classification accuracy. The architecture was proposed by updating the base EfficientNetB0 with different optimizers. However, the EfficientNet models were pre-trained on ImageNet datasets and outperformed other state-of-the-art deep learning models in terms of Top-1 accuracy [52]. This means that the EfficientNetB0 could be the best baseline model since it has low computational complexity, scaling of depth and width, and residual blocks with skip connections to improve the overall classification accuracy [51,52].

### 3.5. BCNet Deep Learning Architecture

The proposed BCNet framework for the human blood cell classification is depicted in Figure 3, while Figure 4 shows the detailed deep learning structures. The base backbone model for deep feature extraction was built based on the EfficientNet-B0 network using seven MBConv blocks, convolution pooling, and fully connected layers [52]. In the proposed BCNet architecture, we adopted and modified the deep learning architecture by adding different layers to concatenate and figure out the extracted deep features and concise them as an average based on the Global Average Pooling (GAP) layer for reducing the occurrence of overfitting [53]. After that, two fully connected layers were added for better class-wise prediction probabilities of all eight classes. The number of neurons per dense layer were determined experimentally to fit our problem of eight blood cell classes using 1024 and eight nodes, respectively. Lastly, the logistic regression function of SoftMax was added to predict the classification scores for each class. Meanwhile, the convolution layer with kernel size of 1 × 1 and number of channels of 1280 was used mainly to decrease the derived deep feature map dimensionality [54]. The local response normalization (LRN) layers were utilized for all MBConv blocks and convolutional layers leading to better prediction performance of the proposed BCNet. However, it was prone to overfitting and lacked the generalization ability of neural architectures. To overcome such a challenge, a dropout strategy with a rate of 0.5 was assigned and used to handle overfitting and BCNet [55]. The dropout rate was essentially used with all convolutional layers as well as dense layers to drop some neural nodes and minimize the number of trainable parameters, reduce the overfitting, and speed-up the learning process [56]. The technical overview of the BCNet architecture is summarized in detail, as demonstrated in Table 1. For training over 5-fold tests, the same training settings and model parameters were used with training/validation sets. The evaluation was achieved via the testing sets. We experimentally fine-tuned the AI models to achieve the best accuracy based on the trail-based error approach [57,58]. All of our experiments are conducted utilizing the various optimization functions available in ADAM, RMSP, and SGDM.

### 3.6. Performance Metrics

The evaluation of the proposed BCNet is carried out for every fold test via a weighted objective metrics strategy including overall accuracy (Az.), F1-score, recall or sensitivity (SE), specificity (SP), Matthews correlation coefficient (MCC), positive predictive value (PPV), and negative predictive value (NPV). The weighted-class strategy is adopted to overcome the class imbalance per test set that contains unbalanced images from different classes. A multi-class confusion matrix is used to derive the evaluation parameters. The evaluation results shown in the result section are achieved over a 5-fold cross-validation test to investigate the reliability and feasibility of the proposed BCNet. The definition of the evaluation metrics is summarized in Equations (1)–(7) [20,59,60,61,62,63]. True positive (TP), true negative (TN), false positive (FP), and false negative (FN) are derived via a multi-class confusion matrix for each fold test. Due to the misbalancing testing images, the weighted evaluation strategy is used to derive all of the evaluation metrics [40].
(1)$$\mathrm{Recall}/\mathrm{Sensitivity}\;\left(\mathrm{SE}\right)=\frac{\mathrm{TP}}{\mathrm{TP}+\mathrm{FN}}.$$
(2)$$\mathrm{Specificity}\;\left(\mathrm{SP}\right)=\frac{\mathrm{TN}}{\mathrm{TN}+\mathrm{FP}}.$$
(3)$$F1-\mathrm{score}\;=\frac{2\cdot \mathrm{TP}}{2\cdot \mathrm{TP}+\mathrm{FP}+\mathrm{FN}}.$$
(4)$$\mathrm{Overall}\;\mathrm{accuracy}\;\left(\mathrm{Az}.\right)=\frac{\mathrm{TP}+\mathrm{TN}}{\mathrm{TP}+\mathrm{FN}+\mathrm{TN}+\mathrm{FP}}.$$
(5)$$\mathrm{MCC}=\frac{\mathrm{TP}\cdot \mathrm{TN}-\mathrm{FP}\cdot \mathrm{FN}}{\sqrt{\left(\mathrm{TP}+\mathrm{FP}\right)\left(\mathrm{TP}+\mathrm{FN}\right)\left(\mathrm{TN}+\mathrm{FP}\right)\left(\mathrm{TN}+\mathrm{FN}\right)}}.$$
(6)$$\mathrm{Precision}/\mathrm{PPV}\;=\frac{\mathrm{TP}}{\mathrm{TP}+\mathrm{FP}}\;.$$
(7)$$\mathrm{NPV}=\frac{\mathrm{TN}}{\mathrm{TN}+\mathrm{FN}}.$$

## 4. Experimental Results

Here, we present and discuss the evaluation classification results obtained using a 5-fold test on the proposed BCNet system. Three optimizers (ADAM, RMSP, and SGDM) are used to conduct comparison evaluations, and the results are recorded and analyzed. Table 2 summarizes the results of the 5-fold test conducted with the cross-validation method applied to the HPBC dataset. The proposed model is a deep learning BCNet model built on the same DL framework, with specific training settings applied to improve the model’s reliability and performance. The classification results with multiple optimizers are reported in Table 1 over 5 folds.

As presented in Table 2, the blood cell classification performance of the proposed BCNet with the RMSP optimizer is the best with almost all fold tests. The average evaluation results over the 5-fold test with all optimizers indicating the error rates are shown in Figure 5. It is shown that the RMSP slightly outperforms other optimizers achieving an overall accuracy of 98.51%, while the ADAM and SGD optimizers achieve 98.47% and 98.24%, respectively. Meanwhile, the BCNet could achieve a promising performance with F1 scores of 96.03%, 96.24%, and 95.30% using ADAM, RMSP, and SGD optimizers, respectively. The evaluation results in terms of the multiclass confusion matrix are depicted for the proposed BCNet with each optimizer, as in the following figures.

As it is shown in all confusion matrices, the low rate of false rates with different classes is recorded. In the case of class Ig (see Figure 6a–c), the majority of the false rate is recorded with the wrongly 48 classified cases as a Neutrophil class, while five and four cases are classified as Monocytes and Basophil, respectively. In the case of the Neutrophil class, the major classification mistake happened as a cross-similarity with the Ig class with 19 images. Such discussion has proven that both Ig and Neutrophil images have similarities in figure, shape, or size, and these comparable nature features could negatively affect the classification performance of any AI-based models. We can also notice that the performance of BCNet with ADAM and RMSP optimizers is much closer to each other since the TP cases are almost the same, whereas, the classification performance with SGD is slightly lower.

## 5. Discussion

Deep learning based on convolutional networks has recently had excellent and successful results in the analysis of medical images for many applications. In this work, we have proposed an AI-based BCNet framework that aims to automatically, accurately, and rapidly recognize multi-class human blood cells, and helps in decision-making for assisting physicians and other medical staff. At the same time, it could assist in the detection of infections based on various blood cell malignancies such as leukaemia, anaemia, etc. Existing studies provided insights on AI-based deep learning for predicting multiple diseases from a specific medical image modality. In our proposed model, we studied a DL-based model for blood cell classification, which consists of eight classes with a publicly available dataset. The proposed model plays a critical role in our model which gets the best detection performance compared with the latest machine and deep learning models. We have used a transfer learning-based approach with the EffieNetB0 model by modifying this model with three optimizers to get the best achievements. In fact, the BCNet is able to detect and classify eight classes: Basophil, Eosinophil, Erythroblast, Ig, Lymphocyte, Monocyte, Neutrophil, and Platelet. To the best of our knowledge, this is the highest number of classes on which a BCNet-based model has been tested in contrast to the existing literature (they used four or five classes). The results of our proposed BCNet, which are presented in Table 2 and Table 3, achieved a competitive overall classification accuracy of 98.51% with an RMSP optimizer and 98.47% with an ADAM optimizer. This study concluded that BCNet with different optimizers achieves promising identification performance with slightly better achievements using the RMSP optimizer. All evaluation results are summarized in the results section.

### 5.1. Comparative Results of the BCNet and Other DL Models

For direct comparison using the same dataset, four deep learning models of DenseNet 201, ResNet50, Inception-V3, and Mobilenet-V2 are adopted and used. These AI models are selected to perform such direct comparison due to their promising classification performance in the research domain [21,37,47,58,59,64,65]. Such comparison is important to investigate the reliability of the proposed model with the trusted ones. The comparison evaluation results among the BCNet and other deep learning models is presented in Figure 7. This result is recorded over 5-fold tests with the same training sets of all models. It is proven that the proposed BCNet could successfully handle and classify all images properly and outperform all other models in terms of all seven metrics.

To explain the rate of change among different models, the standard deviation was measured to check the model diversity with respect to the proposed BCNet model. The performance characteristics regarding this study for all AI models (i.e., DenseNet [1], ResNet [5], Inception [9], and MobileNet [20]) based on the BCNet as a reference are shown in Figure 8. It was clearly indicated that the ResNet had the closest accuracy performance compared with our proposed model, since the prediction error was 0.010. On the other hand, the ResNet showed the worst error deviation in terms PPV and F1-score. The closest model with BCNet was the Inception model in terms of sensitivity and F1-score, with error rates of 0.334 and 0.365, respectively. The DesNet model achieved the best match, with the BCNet achieving the lowest specificity error deviation with 0.030.

The performance comparison of the proposed BCNet against other AI models (i.e., DenseNet, ResNet, Inception, and MobileNet) was carried out in terms of a number of trainable parameters, costs of training time consumption for each epoch, and testing time for a single blood cell image, as shown in Table 3. The proposed BCNet model outperformed the AI models of DenseNet, ResNet, Inception, and MobileNet in terms of the testing time of a single blood cell image by 10.98, 4.26, 2.03, and 0.21 millisecond (msec), respectively. Based on the comparison in Table 3, the heaviest model was the ResNet model, while the MobileNet was comparatively lightweight compared with BCNet. The proposed BCNet model provided promising evaluation performance as well as outperforming the state-of-the-art AI models without regard to the inferencing and testing time cost.

The comparison of the performance of various models with proposed BCNet is carried out overall in terms of several trainable parameters, testing time for each image, and costs of training time consumption for each epoch with models such as DenseNet, ResNet, Inception, MobileNet and Proposed BCNet. The proposed model, with less training and testing costs, has shown promising results.

### 5.2. Comparison between Proposed BCNet and Previously Published Models

A comparison between our proposed BCNet against the latest deep learning works for blood cell classification is performed and summarized in Table 4. The proposed BCNet could provide a promising and competitive evaluation result for multiple classes.

### 5.3. Ablation Study

We performed this ablation study to show the proposed model capability in improving the overall classification evaluation performance with/without the additional layers on the base model. To achieve this study, the comparison evaluation results against the baseline model were conducted and the derived evaluation results are reported in Table 5. We retrained and evaluated the proposed model against its base model using three optimizers: Adam, RMSP, and SGD. It was clearly shown that the prediction accuracy performance was improved due to our modifications by 1.94%, 3.33%, and 1.65% in terms of using ADAM, RMSP, and SGD optimizers, respectively. Similarly, the performance was improved in terms of F1-scores by 1.14%, 2.13%, and 0.75% with respect to the ADAM, RMSP, and SGD optimizers, respectively.

### 5.4. Limitations and Future Work

The most significant limitation of the AI-based medical application is due to the scarcity of the annotated benchmark datasets since the labeling process is expensive and it takes time for the experts to do it. In the future, we plan to check the capability of the BCNet with other medical imaging such as magnetic resonance imaging (MRI), Ultrasound (US), X-ray, and so on. Furthermore, we plan to continue improving the performance behavior and providing more interesting blood cell identification performance by applying the newly impressive AI technologies such as explainable AI, federated learning, etc.

## 6. Conclusions

One of the most significant issues that the field of medicine is currently facing is the classification of blood cells. This is especially true given the growing number of infection cases and the difficulty in identifying them at an early stage. Within the scope of this work, we have investigated the classification of blood cells by utilizing transfer deep learning, BCNet. An exhaustive experimental investigation is carried out with several different optimizers in order to assess the dependability and practicability of the suggested model. At the same time, we examine the similarities and differences between a variety of transfer learning models and various optimizers. The RMSP optimizer was used to produce the best possible accuracy of 98.51%, which exceeds other deep learning models that are currently available. When applied to the classification of blood cells into eight different classes, our suggested method achieves a high level of accuracy. In order to demonstrate that the transfer learning model with the RMSP activation function is an effective one, we compared this study to a large number of other similar works.

## Figures and Tables

**Figure 1 diagnostics-12-02815-f001:**
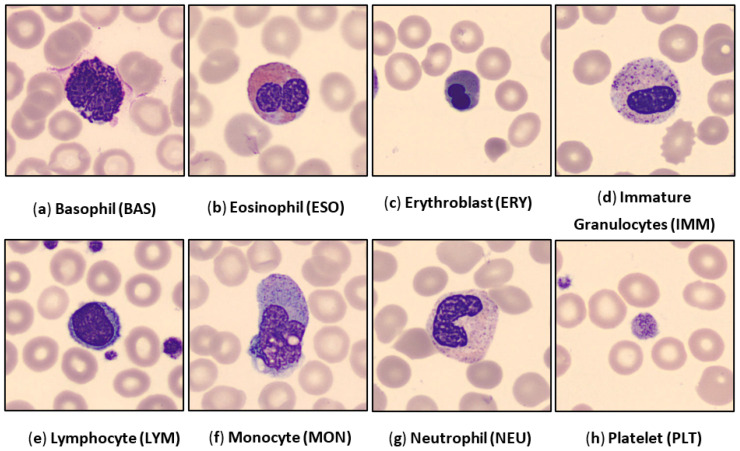
Examples of the human peripheral blood cell (HPBC) images over all eight classes.

**Figure 2 diagnostics-12-02815-f002:**
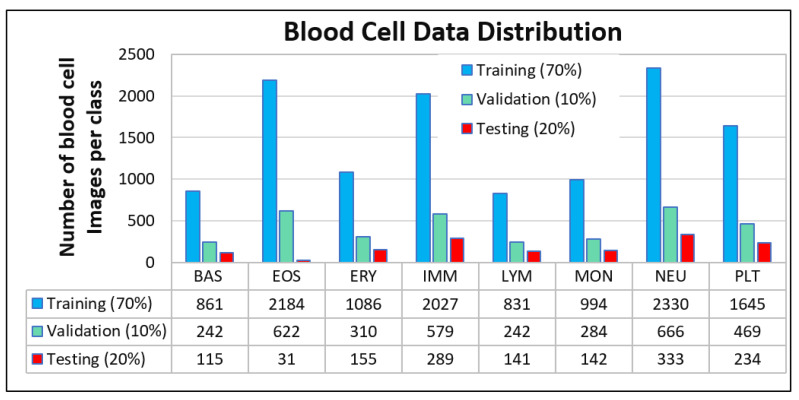
HPBC dataset distribution over eight different classes of the human peripheral blood cell (HPBC) images. The images were randomly split per class into 70% (training), 10% (validation), and 20% (testing).

**Figure 3 diagnostics-12-02815-f003:**
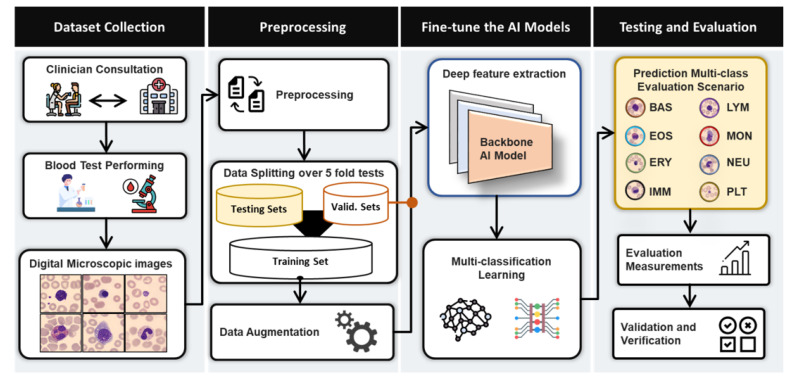
Abstract view paradigm of the proposed BCNet for blood cell multi-class identifications.

**Figure 4 diagnostics-12-02815-f004:**
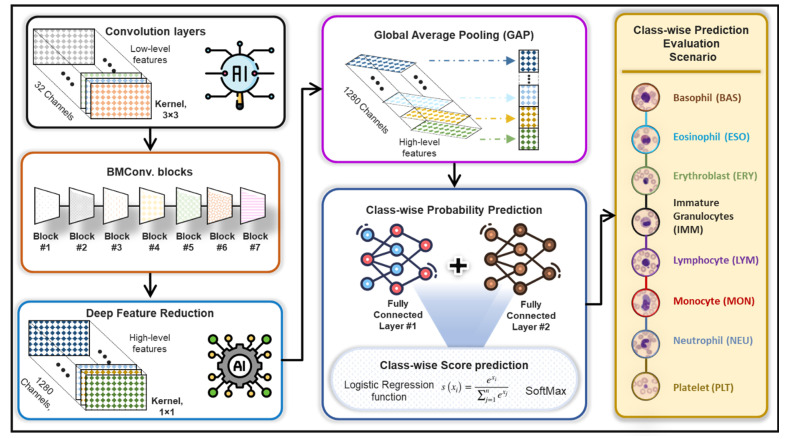
Feature extraction methodology of the proposed BCNet framework.

**Figure 5 diagnostics-12-02815-f005:**
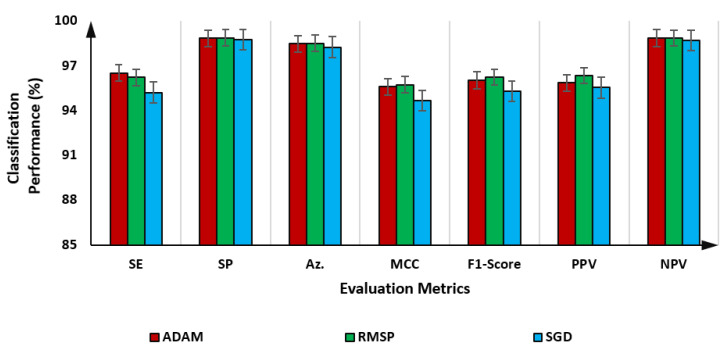
Evaluation classification results of the proposed BCNet as an average over 5-fold tests using ADAM, RMSP, and SGD optimizers.

**Figure 6 diagnostics-12-02815-f006:**
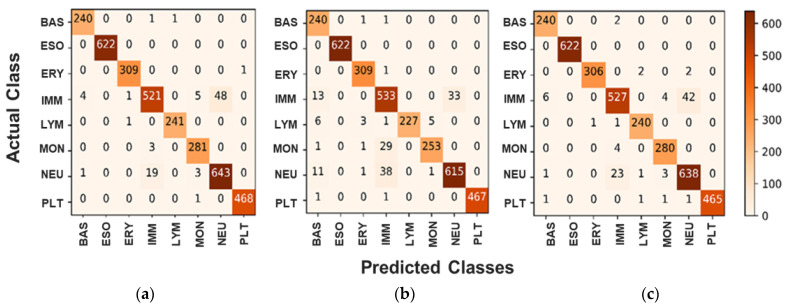
Confusion Matrix for Multiclass scenario for the proposed BCNet with different optimizers. (**a**) ADAM optimizer, (**b**) RMSP optimizer, and (**c**) SGD optimizer.

**Figure 7 diagnostics-12-02815-f007:**
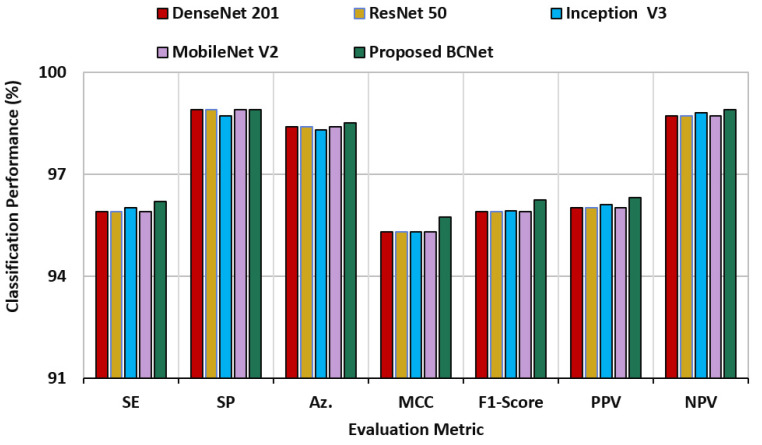
Direct comparison results of blood cell classification using the proposed BCNet and other deep learning models of DenseNet 201, ResNet50, Inception-V3, and Mobilenet-V2.

**Figure 8 diagnostics-12-02815-f008:**
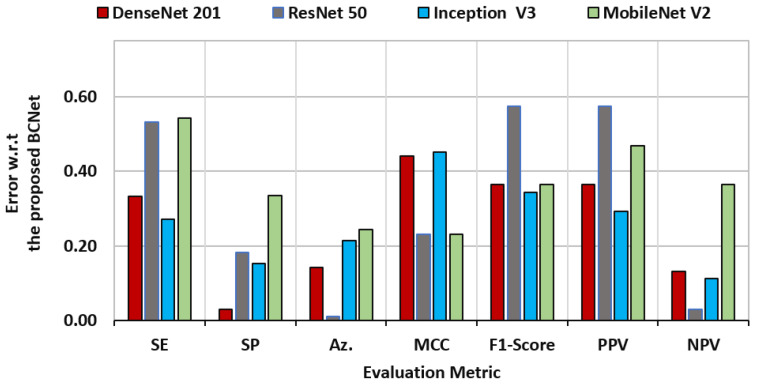
Evaluation error performance of the latest deep learning models of DenseNet 201, ResNet 50, Inception V3, and Mobile Net V2 with respect of the proposed BCNet model.

**Table 1 diagnostics-12-02815-t001:** Technical Parameters of the proposed BCNet network.

Stage	Operatory	Spatial Resolution Hi × Wi	Channel, Ci	Layer, Li
1	Conv., k3 × 3	224 × 224	32	1
2	MBConv1, k3 × 3	112 × 112	16	1
3	MBConv6, k3 × 3	112 × 112	24	2
4	MBConv6, k3 × 3	56 × 56	40	2
5	MBConv6, k3 × 3	28 × 28	80	3
6	MBConv6, k3 × 3	14 × 14	112	3
7	MBConv6, k3 × 3	14 × 14	192	4
8	MBConv6, k3 × 3	7 × 7	320	1
9	Conv1 × 1,	7 × 7	1280	1
10	GAP			
11	2 Dense Layers		8 nodes	
12	SoftMax		8 nodes: Number of blood cell classes.	

Conv: Convolution neural network. MBConv: Mobile Inverted convolution blocks. FC: Fully connected. GAP: Global average pooling.

**Table 2 diagnostics-12-02815-t002:** Evaluation classification results (%) of the proposed BCNet with different optimizers for blood cell classification over 5-fold tests.

No. of Fold	Optimizer	SE	SP	Az.	MCC	F1-Score	PPV	NPV
Fold 1	ADAM	93.89	98.14	97.53	90.93	91.66	90.44	98.55
	RMSP	95.53	98.9	98.47	95.11	95.61	95.8	98.84
	SGD	93.12	98.53	97.83	92.55	93.36	93.89	98.47
Fold 2	ADAM	97.91	99.26	99.04	97.74	97.74	98.02	99.12
	RMSP	98.3	99.26	99.13	98.15	98.3	98.33	99.22
	SGD	93.12	98.53	97.83	92.55	93.36	93.89	98.47
Fold 3	ADAM	96.8	98.88	98.53	96.28	96.79	96.82	98.87
	RMSP	95.09	98.64	98.13	94.37	95.1	95.21	98.63
	SGD	96.6	98.85	98.49	96.05	96.59	96.61	98.84
Fold 4	ADAM	97.16	98.99	98.7	96.73	97.15	97.15	98.86
	RMSP	97.1	98.96	98.67	96.66	97.09	97.12	98.96
	SGD	96.63	98.93	98.57	96.13	96.62	96.63	98.88
Fold 5	ADAM	96.8	98.88	98.53	96.28	96.79	96.82	98.87
	RMSP	95.09	98.64	98.13	94.37	95.1	95.21	98.63
	SGD	96.6	98.85	98.49	96.05	96.59	96.61	98.84
Avg. (%)	ADAM	96.51	98.83	98.47	95.59	96.03	95.85	98.85
	RMSP	96.22	98.88	98.51	95.73	96.24	96.33	98.86
	SGD	95.21	98.74	98.24	94.67	95.30	95.53	98.70

**Table 3 diagnostics-12-02815-t003:** Performance comparison of the proposed BCNet against the latest AI models with respect to the number of the learning parameters and the computation costs.

AI Model	Number of Trainable Parameters (million)	Training Time Per Epoch (sec.)	Testing Time/Image (msec.)
DenseNet 201	18.10	286	18.13
ResNet 50	23.55	148	11.41
Inception V3	21.78	128	9.18
MobileNet V2	2.23	123	7.36
The proposed BCNet	4.017	122	7.15

**Table 4 diagnostics-12-02815-t004:** Evaluation comparison results for blood cell identification via the proposed BCNet against the latest deep learning works in the literature.

Reference	Data	Methods	Az. (%)
Zhao et al., 2017 [44]	Cell vision, ALL-IDB, Jiashan	CNN, SVM, and random forest	92.80
Journal et al., 2021 [66]	Collected, BCCD data set	Two DCNN	95.17 (Precession)
Acevedo et al., 2019 [32]	Private	CNN + Transfer learning	96
Qin et al., 2018 [29]	Private	CNN	76.84
Ma et al., 2020 [40]	BCCD	DCGAN + Transfer learning	91.7
Baydilli and Atila 2020, [67]	LISC	Capsule network	96.86
Rui Liu et al., 2022 [37]	HPBC	Transfer Learning	96.83
The proposed BCNet	HPBC images	BCNet	98.51

**Table 5 diagnostics-12-02815-t005:** The compression evaluation result of the ablation study.

AI Models		SE	SP	Az.	MCC	F1-Score	PPV	NPV
Baseline Model	ADAM	95.8	97.88	96.53	93.38	94.89	94.82	96.88
	RMSP	94.09	95.66	95.18	94.37	94.11	94.21	95.63
	SGD	93.50	96.65	96.59	95.05	94.55	95.61	97.84
The Proposed BCNet	ADAM	96.51	98.83	98.47	95.59	96.03	95.85	98.85
	RMSP	96.22	98.88	98.51	95.73	96.24	96.33	98.86
	SGD	95.21	98.74	98.24	94.67	95.30	95.53	98.70

## Data Availability

The dataset used in this study is publicly available via https://data.mendeley.com/datasets/snkd93bnjr/1 (accessed on 21 June 2022).

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
