# Peer review of "BCNet: A Deep Learning Computer-Aided Diagnosis Framework for Human Peripheral Blood Cell Identification"

_diagnostics, 2022, doi:10.3390/diagnostics12112815_

Round 1

Reviewer 1 Report

1. A thorough check of language, and typos are essential. Update figure 3  label

2. Remove repetition of F1 score equation 3. Even thoroughly check the relationship in equations 1 -6 and eliminate errors. Equation 5 is expressed as   MMC ( check it), and in the table specified as MCC. 

3. Clearly mention the data source.

4. How the proposed method improves results significantly to 98.51%

5. Authors need to cite a few more latest papers in the domain

6. Is cross-validation is performed? If so mention the K value.

7. Table 1 needs a  careful check.

8. Enhance the result section with additional accuracy  and error curve

Author Response

  • First of all, we would like to thank expert reviewer#1 for his insightful comments and suggestions to improve the overall manuscript's annovation and structure. Really, the reviewer's comments improve the manuscript and give it more highlight availbailty for other researchers. Thanks again!
  • Here, we have addressed the all comments raised by the reviewer and tried our best to answer them one-by-one. Accordingly, the manuscript is revised based on the reviewer's suggestions and recommendations. 
  • Please, find the attached PDF file that contains the response answers to all reviewer's comments. 

Reviewer 2 Report

This paper proposes BCNet neural network to classify the blood cells in eight-class identification scenarios. The topic is interesting and matches well for Diagnostics journal. The paper contains some review of related works. However the paper has the following major concerns.

The authors describe the major contribution of the paper as follows:

«● A deep learning schema of the CAD system based on the newly deep learning BCNet is proposed in order to identify multiclass blood cells rapidly and automatically.

The multiple class identification task is conducted in terms of improving the overall classification performance.

A comprehensive evaluation experiment is conducted to investigate the reliability and feasibility of the proposed BCNet using multiple optimizers and different state-of-the-art deep learning models such as DensNet, ResNet, Inception, and MobileNet.»

1. The authors used the following dataset:

«For developing and evaluating the deep learning BCNet and other models, a public dataset of human peripheral blood cells (HPBC) is used. The cell images are collected, in the period of 2015 to 2019, from the hospital clinic of Barcelona and prepared by Anna et al. [21].»

By reference 21, I could not find either the author Anna or the dataset. In my opinion, the authors need to take a public dataset that has already been used in other publications to solve the same problem. And compare their results with the results of other authors on the same data set.

2. The authors compare their proposed neural network with a random set of other neural networks for classification. Such a comparison does not make sense if you do not specify the parameters of these neural networks - the number of weights, the inference time. Comparison with other neural networks could be more systematic if the authors compared the neural network proposed by them based on EfficientNet-B0, for example, with other neural networks from the same family - B1...B7. With an indication, respectively, of the number of weights, the time of inference.

3. The BSNet neural network proposed by the authors is a modification of the EfficientNet-B0 basic neural network. Authors must complete the ablation study, i.e. show that their modifications led to an increase in classification accuracy.

4. There are also typos in the paper.

- «Of these deaths, about 350 will be attributable to lung cancer» — 350,000?

- «Mopilenet-V2»

Author Response

  • First of all, we would like to thank expert reviewer#1 for his insightful comments and suggestions to improve the overall manuscript innovation and structure. Really, the reviewer's comments improve the manuscript and give it more highlight availability for other researchers. Thanks again!
  • Here, we have addressed the all comments raised by the reviewer and tried our best to answer them one-by-one. Accordingly, the manuscript is revised based on the reviewer's suggestions and recommendations. 
  • Please, find the attached PDF file that contains the response answers to all reviewer's comments. 

Round 2

Reviewer 2 Report

It is OK now.